# Absence of Anti-*Babesia microti* antibody in commercial intravenous immunoglobulin (IVIG)

Julia Kostka[1]☯*, Anu S. Maharjan[2]☯, Sanjai Kumar[3], Douglas Hackenyos[4], Peter J. Krause[5]‡, Kevin Dieckhaus[1]‡

1 Infectious Disease, UConn Health, Farmington, Connecticut, United States of America, 2 Pathology and Laboratory Medicine, UConn Health, Farmington, Connecticut, United States of America, 3 Food and Drug Administration, Silver Spring, Maryland, United States of America, 4 Pharmacy, UConn Health, Farmington, Connecticut, United States of America, 5 Yale School of Public Health and Yale School of Medicine, New Haven, Connecticut, United States of America

☯ These authors contributed equally to this work.
‡ These authors are co-last authors on this work.
* kostka@uchc.edu

**Data Availability Statement:** All relevant data are within the manuscript and its Supporting Information files.

## Abstract

### Background

Babesiosis is a worldwide emerging protozoan infection that is associated with a spectrum of disease severity from asymptomatic infection to severe organ damage and death. While effective treatment strategies are available, some immunocompromised patients experience severe acute and prolonged/relapsing illness due in part to an impaired host antibody response. Intravenous immunoglobulin (IVIG) has been used as an adjunctive therapy in some immunocompromised babesiosis patients, but its therapeutic effect is uncertain. We evaluated the presence of *Babesia microti* antibodies in commercial samples of IVIG.

### Methods/Principle findings

The presence of *B. microti* antibodies in commercial samples of IVIG were tested using an immunofluorescence assay. A subset of samples was then tested for *B. microti* antibodies using an enzyme linked immunosorbent assay.

Out of 57 commercial IVIG samples tested using IFA, and 52 samples tested using ELISA, none were positive for *B. microti* antibodies.

### Conclusions

Commercially available IVIG may not be of therapeutic benefit for babesiosis patients. Additional sampling of IVIG for B. microti antibody and a clinical trial of babesiosis patients given IVIG compared with controls would provide further insight into the use of IVIG for the treatment of babesiosis.

**Funding:** This work was supported by UCONN Health, division of infectious disease and was supported in part by a gift from the The Llura A. Gund Laboratory for Vector- borne Disease Research and the Gordon and Llura Gund Foundation (PJK). It was in part funded by the FDA Intramural Research Program. This article reflects the views of the author and should not be construed to represent FDA's views or policies. The funding organizations played no role in study design, data collection and analysis, decision to publish or preparation of the manuscript. Founder websites can be found here: https://health.uconn.edu/infectious-diseases/ https://fconline.foundationcenter.org/fdo-grantmaker-profile/?key=GUND012 https://www.fda.gov/emergency-preparedness-and-response/mcm-regulatory-science/intramural-research.

**Competing interests:** The authors have declared that no competing interests exist.

## Author summary

Human babesiosis is usually a mild to moderate illness that clears with a standard course of atovaquone and azithromycin. The disease is often severe in immunocompromised patients, including those over 50 years of age, those with asplenia, malignancy, HIV/AIDS, or on immunosuppressive therapy. Patients with multiple immunosuppressive conditions that usually result in low or absent antibody production may experience severe acute disease followed by persistent relapsing disease that can last for months or even years. Intravenous immunoglobulin (IVIG) has been used as adjunct therapy in patients with severe babesiosis, although controlled therapeutic trials have not been carried out. This prompted us to evaluate *Babesia microti* antibody concentrations in IVIG samples from different commercial sources. We tested commercially available IVIG for antibodies against *B. microti* using IFA and ELISA assays. None of the IVIG samples contained detectable antibody against B. microti. Our results suggest that commercially available IVIG may be of limited therapeutic benefit for babesiosis patients.

## Introduction

Babesiosis is a worldwide emerging protozoan infection that is endemic in the northeastern and northern Midwestern United States. A total of 16,456 US cases were reported to the CDC between 2011–2019 [1]. The disease is caused by a number of *Babesia* spp., with *Babesia microti* the most common cause of human infection [2]. *B. microti* is transmitted to humans via *Ixodid* (hard bodied) ticks, primarily by *Ixodes scapularis* in the United States. Humans also rarely may become infected with *Babesia* spp. through blood transfusion, organ transplantation, or congenital transmission [2,3]. Symptoms typically include fever, fatigue, chills, sweats, anorexia, headache, myalgias, and malaise. Severe disease requires hospital admission and can lead to multiorgan complications and death [2]. Prompt diagnosis is important, especially in immunocompromised patients, and consists of identification of *Babesia* spp. on peripheral blood smear(s) and/or PCR testing. Current treatment recommendations include a combination of azithromycin and atovaquone, with clindamycin and quinine being second-line therapy [4]. Exchange transfusion may be used in severe disease with parasitemia >10% and/or patients with end-organ impairment [5].

Most babesiosis patients respond well to atovaquone and azithromycin but some highly immunocompromised patients experience relapsing illness despite prolonged antimicrobial therapy [6]. The majority of these patients have B-cell lymphoma or autoimmune disease and have been treated with rituximab, a monoclonal antibody that targets B cells [6,7]. These patients often have low plasma anti-*B. microti* antibody concentrations. Those immunosuppressed patients who have difficulty clearing the infection require prolonged therapy and this leads to the emergence of antimicrobial resistant strains that are even more difficult to eradicate.

Rituximab, an anti-CD20 monoclonal antibody is used predominantly in non-Hodgkin lymphoma (NHL) and autoimmune conditions [8,9]. It was the first monoclonal antibody (mAb) used for oncology patients and is typically used in combination with chemotherapeutic agents but can also be used as monotherapy [8]. Rituximab leads to B-cell depletion and decreased production of antibody within 72-hours and lasts for approximately six to 12 months [8,10–11]. During this time, patients have diminished antibody and are at increased risk of infections.

Consequently, intravenous immunoglobulin (IVIG) has been used to provide anti-*B. microti* antibody even though the concentration of *B. microti* antibody in IVIG and its therapeutic efficacy are unknown. We treated a babesiosis patient with IVIG with uncertain benefit, prompting us to evaluate *B. microti* antibody concentrations in IVIG samples from different commercial sources.

## Methods

### Ethics statement

Human subjects research determinations were sought from the UConn Health Institutional Review Board and the United State Food and Drug Administration. The project was determined by each to not constitute human subjects research, therefore there is no associated approval number.

**Case report.** A 66-year-old male patient with B cell lymphoma developed fever and was diagnosed with babesiosis using PCR. He received a total of six cycles of EPOCH therapy, including Rituximab, every three weeks. His last dose was administered approximately four months prior to the development of babesiosis. The initial babesiosis symptoms resolved after a 10-day course of azithromycin (1000 mg daily) and atovaquone (750 mg twice daily). They recurred approximately two months later and a blood smear revealed 0.33% parasitemia while *B. microti* PCR remained positive. He was again placed on azithromycin 1000 mg daily and atovaquone 750 mg twice daily but 10 weeks later continued to exhibit persistent symptoms and low level parasitemia. Clindamycin (600 mg every 8 hours) was added to his regimen. A negative IFA titer to *Babesia spp*, absence of B cells on peripheral blood flow cytometry and low plasma levels of IgG1 and IgG2 were noted at that time (Fig 1). Accordingly, IVIG (Privigen, 65g/dose given monthly) was initiated after written informed consent was obtained per hospital protocol, in addition to clindamycin (600 mg every 8 hours), azithromycin (1000 mg daily), atovaquone (750 mg twice daily). Three months later, symptoms had resolved, blood smears were negative, and an IFA IgG titer to *B. microti* was positive at 1:256 (Fig 1). *B. microti* PCR subsequently reverted to negative after 8 months of therapy, coinciding with recovery of B cells as demonstrated by peripheral flow cytometry. The patient received an additional 3

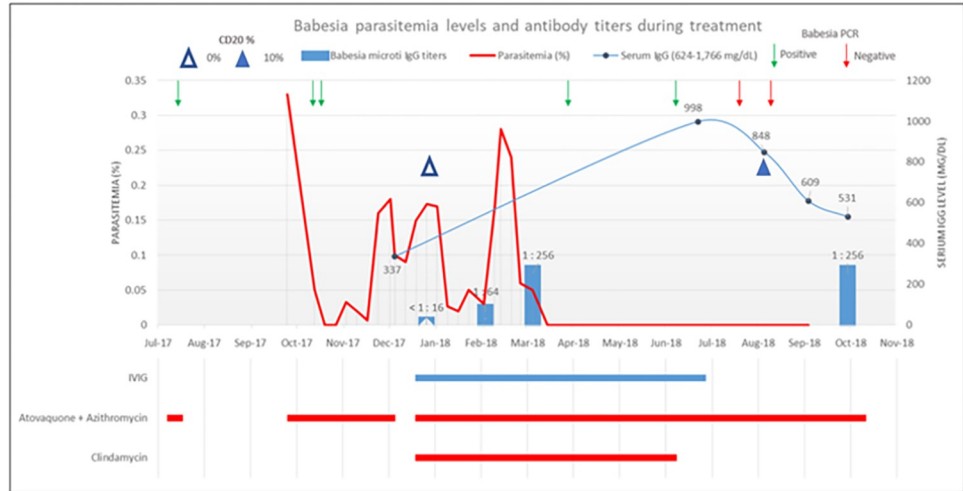

**Fig 1. *Babesia* parasitemia levels and antibody titers during treatment.** Timeline of *Babesia microti* parasitemia, amplifiable *B. microti* DNA in blood by PCR, percentage of CD20 cells (B cells), and antibody level in an immunocompromised host.

months of azithromycin and atovaquone and has not experienced a clinical relapse over the past five years. In order to determine whether IVIG might have contributed to the clearance of infection, we evaluated 57 lots of IVIG from 4 commercial products for *B. microti* antibody using indirect fluorescent antibody (IFA) and enzyme linked immunosorbent assay (ELISA).

**IVIG preparations:** Unused IVIG obtained from clinical infusions at the University of Connecticut Infusion Clinic were evaluated in this study (S1 Table). After dose preparation and clinical infusion, tubing and product bottles were allowed to settle for approximately 20 minutes to allow pooling of liquid IVIG in the discard materials. Remnant liquid was drawn off and stored in ~0.5 ml aliquots at -80°C. Each patient infusion typically yielded 1 to 3 aliquots for subsequent antibody testing.

### *Babesia microti* immunofluorescence assay

We first determined the presence of *B. microti* antibody using a modification of the standard *B. microti* immunofluorescence assay (IFA) [8,9]. *Babesia microti*–infected red blood cells (iRBCs) grown in BALB/cJ mice were used as the antigen source. Slides were prepared using a diluted suspension of $5 \times 10^6$ *B. microti* iRBCs/10μL, dried overnight, and stored at –70°C until use. Test IVIG samples were then diluted to 1:64 and 1:128 in 1× PBS/bovine serum albumin and added in 20-μL increments to each well on a 12-well slide. Two positive and one negative control sera were included for each test. Slides were processed as previously described [12,13]. A total of 57 samples were tested including 34 Privigen, 19 Gammagard S/D, 2 Gamunex-C, and 2 Gammagard Liquid.

### *Babesia microti* ELISA assay

In order to further determine the presence or absence of *B. microti* antibody, a subset of samples were tested using an ELISA (InBios International, Inc., Seattle, WA). A total of 52 samples were tested, including 30 Privigen, 18 Gammagard S/D, 2 Gammagard Liquid and 2 Gamunex-C. Assuming a physiological extracellular concentration of IVIG at 8.8 mg/mL in a 70 kg (800 mg/kg) patient, IVIG samples were diluted to 1:32, 1:64, and 1:128 concentrations for antibody testing. To test the 1:32 dilution, 82.5 μL of 100 mg/mL IVIG was aliquoted into 217.5 μL of human serum. Four μL of this dilution mixture was then aliquoted into 396 μL of ELISA sample dilution buffer as indicated in the manufacturer's instructions (InBios International, Inc., Seattle, WA). A negative control without IVIG was prepared by aliquoting 82.5 μL of 0.9% saline (Baxter, Deerfield, IL) with 217.5 μL of human serum (Sigma, St. Louis, MO). For 1:64 and 1:128 dilutions, 41.4 μL and 20.6 μL of 100 mg/mL IVIG were aliquoted into 258.6 μL and 279.4 μL of human serum, respectively. These dilution mixtures were then further diluted with ELISA sample dilution buffer as described above. Control samples consisted of no IVIG control (saline with human sera; Sigma), the manufacturers' positive and negative controls, and serum from one patient with previously documented *B. microti* infection (undiluted, 1:64 and 1:128 dilutions). Three different concentrations of IVIG, a no IVIG control, and a *B. microti* positive patient serum sample were prepared as described above. The undiluted patient positive control and manufacturer positive control were run in duplicate. Samples were then incubated in *B. microti* coated recombinant protein-based ELISA plate wells according to the manufacturers' instructions. The procedure for incubation, washing, and detection were all carried out as described in the manufacturer's instructions. The manufacturer defines an OD450 ≤ 0.30 as negative and ≥ 0.50 as positive.

## Results

### IFA

We measured the anti-*B. microti* IgG titers against blood-form parasites in the IVIG samples using infected mouse red cells as antigen source by IFA. We first determined the optimal IgG concentration in the IVIG lots that would not give a non-specific reactivity against mouse RBC in IFA. We found that IgG concentration of $\leq$ 200 µg/ml allowed a clear *B. microti* parasite staining without a background signal. Therefore, each IVIG sample was tested at four IgG concentrations of 200 µg/ml, 100 µg/ml, 50 µg/ml, and 25 µg/ml by IFA. Results showed none of the IVIG samples tested positive by IFA.

### ELISA

Using the ELISA manufacturer's cutoff values for positive and negative results, the manufacturer positive controls and patient sample positive controls were positive, while the negative controls samples tested negative for *B. microti* antibodies (Fig 2). Each of the IVIG samples from different manufacturers diluted in serum at 1:32 (n = 26), 1:64 (n = 52), and 1:128 (n = 52) tested negative for *B. microti* antibodies (Fig 2). Twenty-one of the IVIG samples diluted 1:32 had an OD450 of <0.30, and five had an OD450 of <0.50. For those diluted at 1:64, forty-nine had an OD450 of <0.30, and three had an OD450 of <0.50. All 52 samples diluted at 1:128 had an OD450 of <0.30.

## Discussion

To our knowledge, this is the first study that has assessed the *B. microti* antibody content of IVIG. IVIG is derived from pooled plasma donors [14]. It provides passive immunity and has

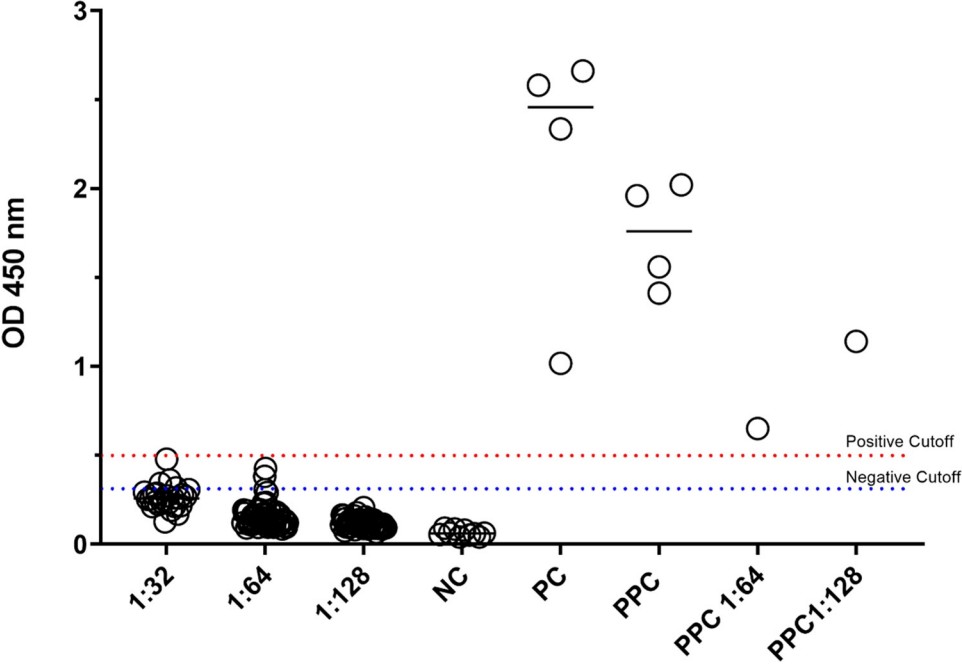

**Fig 2. *Babesia microti* antibody titers by ELISA in IVIG.** *Babesia microti* antibody concentrations in IVIG and human serum samples using InBios ELISA kit. Positive human serum specimens and 28 IVIG products from three commercial sources were diluted to 1:32 and 52 IVIG products were further diluted to 1:64, and 1:128. The dashed blue line represents the negative cutoff of $OD_{450}$ 0.300 and the red line represents the positive cutoff of $OD_{450}$ 0.500. PC: manufacturer positive *B. microti* antibody control; NC: no IVIG control; PPC: positive human serum *B. microti* antibody control.

been used for various infectious, autoimmune, and immunodeficiency disorders with varying success [14]. For example, IVIG is effective treatment for Guillain Barre syndrome, ITP, and Kawasaki's disease, but has inconsistent effect on chronic hepatitis B. It is occasionally used for treatment of patients experiencing persistent babesiosis but there are no clinical trial studies that demonstrate efficacy. Antibody is thought to play an important role in clearing *B. microti* infection because persistently infected patients have B cell lymphoma and/or rituximab therapy or other conditions with low or absent *B. microti* antibody blood concentrations [6,15]. The Fab region of rituximab binds to a sequence of CD20 on B cells that is four amino acids in length [9]. In so doing, it depletes B-cells and the production of antibody, although it is thought that there are other mechanisms by which rituximab leads to B-cell depletion [9]. Rituximab-induced signaling may act synergistically with chemotherapeutics to induce cell death [16]. Furthermore, rituximab leads to complement mediated cytotoxicity and rapid B cell death [16]. Antibody-dependent cellular cytotoxicity is induced by rituximab through effector cells [16,10]. As such, patients who received rituximab may be unable to mount an effective antibody response to *B. microti*.

IVIG might be expected to have at least some anti-*B. microti* antibody if plasma donors were residents of *Babesia*-endemic areas. Roughly 4–14% of plasma collection sites from IVIG companies used in our study (Grifols/Gamunex-C, CSL Plasma/Privigen, BioLife/Gammagard) are located in *B. microti*-endemic areas (Table 1) [17–19]. Even in highly endemic locales, only 1.1–1.4% of blood donors are *B. microti* antibody positive [20]. In broader general populations, *B. microti* antibody varies but can be as high as 10% [13,21]. Although the exact degree of *B. microti* antibody-positive donors associated with the IVIG samples is unknown, it is likely to be low based on the location of collection sites in both endemic and non-endemic areas, and low baseline seropositivity found in blood donors in endemic sites. Furthermore, pooling of plasma samples in the large IVIG lots would significantly dilute antibody level from donor(s) who may have been positive for *B. microti* antibody. None of the IVIG samples that we tested were positive for *B. microti* antibody against whole parasite by IFA or against recombinantly produced antigens by ELISA, suggesting that commercially available IVIG may not be effective therapy for babesiosis patients.

Despite our findings, anti-*B.microti* antibodies (1:64 titer) appeared within a month of initiation of IVIG and increased (1:256 titer) over the following month. A repeat antibody test 7 months later was also positive (1:256 titer). Although commercial lots of IVIG appear to lack *B. microti* antibody, IVIG may act indirectly to stimulate antibody production. Low doses of IVIG have been shown to induce proliferation and immunoglobulin synthesis in Common Variable Immunodeficiency patients [22]. In contrast, IVIG may lead to B-cell apoptosis via interaction with the CD22-receptor protein on the B cell membrane, which disrupts cell signaling pathways [23,24]. IVIG has been shown to down-regulate CD21, which also increases B-cell apoptosis [24]. Overall, at least five IVIG-induced mechanisms have been described to affect B-cells, including (i) sustained activation of Erk1/2 (extracellular kinase) which leads to apoptosis, (ii) inhibition of B-cell receptor-induced calcium signals and cellular proliferation,

**Table 1. Number and percent of plasma collection sites located in *Babesia microti*-endemic states in the US.**

| Study sites | Grifols (Gamunex-C) | CSL Plasma (Privigen) | BioLife (Gammagard Liquid & Gammagard S/D) |
|---|---|---|---|
| Endemic area collection sites* | 12 | 35 | 30 |
| Total Collection Sites | 291 | 321 | 216 |
| Percentage sites in endemic areas | 4 | 11 | 14 |

*Connecticut, Maine, Massachusetts, Minnesota, New Hampshire, New Jersey, New York, Rhode Island, Vermont, Wisconsin

(iii) anti-idiotypic binding of IVIG that modulates the production of pathogenic autoantibody, (iv) B-cell activation, and (v) suppression of proinflammatory cytokine production [24]. It is also possible that the rise in antibody at the time of IVIG infusion was due to a decreased rituximab effect rather than IVIG infusion. The emergence of antibody occurred 9 months after the last dose of rituximab and coincides with the expected timing of B-cell recovery [10–11]. In addition, IVIG was administered concurrently with the addition of clindamycin to azithromycin and atovaquone. A three-drug regimen is one of the therapeutic measures recommended by experts in immunocompromised patients who fail to respond to initial therapy and may have also contributed to the patient's recovery [4].

In conclusion, our data raises uncertainty whether commercially available IVIG may not be an effective adjunct therapy in immunosuppressed patients who responded poorly to anti-*Babesia* treatment. None of the IVIG samples that we tested contained *B. microti* antibodies as measured by IFA (n = 57) or ELISA (n = 52). Clinical studies using IVIG products specifically sourced from donors screened for *B. microti* antibody may provide further insight into their potential therapeutic benefits. Additionally, monoclonal antibodies directed against immunodominant *B. microti* proteins with proven protective efficacy in preclinical studies may be an attractive alternative in treatment of *B. microti*-infected immunocompromised patients, but clinical trials are needed to confirm this possibility [25].

## Supporting information

**S1 Table. Commercial IVIG Samples.** Table with listed commercial IVIG samples used for ELISA and IFA testing, including collection date, product, lot number and ELISA results(s).
(DOCX)

**S2 Table. Numerical values corresponding to Fig 1.** Table with values of *Babesia microti* IgG, serum IgG, CD 20% and parasitemia %.
(DOCX)

## Author Contributions

**Conceptualization:** Peter J. Krause, Kevin Dieckhaus.

**Investigation:** Julia Kostka, Anu S. Maharjan, Sanjai Kumar.

**Resources:** Douglas Hackenyos, Peter J. Krause, Kevin Dieckhaus.

**Writing – original draft:** Julia Kostka, Anu S. Maharjan.

**Writing – review & editing:** Sanjai Kumar, Douglas Hackenyos, Peter J. Krause, Kevin Dieckhaus.

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
