## [Decision Letter · Decision Letter 0]

20 Nov 2023

Dear Dr. Kostka,

Thank you very much for submitting your manuscript "Absence of Anti-<i>Babesia microti<i> Antibody in Commercial Intravenous Immunoglobulin (IVIG)" for consideration at PLOS Neglected Tropical Diseases. As with all papers reviewed by the journal, your manuscript was reviewed by members of the editorial board and by several independent reviewers. In light of the reviews (below this email), we would like to invite the resubmission of a significantly-revised version that takes into account the reviewers' comments. 

We cannot make any decision about publication until we have seen the revised manuscript and your response to the reviewers' comments. Your revised manuscript is also likely to be sent to reviewers for further evaluation.

Sincerely,

Angela Monica Ionica, Ph.D.

Academic Editor

Abhay Satoskar

Section Editor

Reviewer's Responses to Questions

**Key Review Criteria Required for Acceptance?**

**Methods**

-Are the objectives of the study clearly articulated with a clear testable hypothesis stated?

-Is the study design appropriate to address the stated objectives?

-Is the population clearly described and appropriate for the hypothesis being tested?

-Is the sample size sufficient to ensure adequate power to address the hypothesis being tested?

-Were correct statistical analysis used to support conclusions?

-Are there concerns about ethical or regulatory requirements being met?

Reviewer #1: all aspects are acceptable.

Reviewer #2: The study was designated appropriate and all analysis were performed correctly. I have one comment to clarify – why were only a portion of IVIG samples tested using ELISA, while 57 commercial samples were tested using IFA?

Reviewer #3: (No Response)

**Results**

-Does the analysis presented match the analysis plan?

-Are the results clearly and completely presented?

-Are the figures (Tables, Images) of sufficient quality for clarity?

Reviewer #1: No problem.

Reviewer #2: I have no comments to Result section.

Reviewer #3: (No Response)

**Conclusions**

-Are the conclusions supported by the data presented?

-Are the limitations of analysis clearly described?

-Do the authors discuss how these data can be helpful to advance our understanding of the topic under study?

-Is public health relevance addressed?

Reviewer #1: It is ok

Reviewer #2: I have no comments to Conclusions section.

Reviewer #3: (No Response)

**Editorial and Data Presentation Modifications?**

Reviewer #1: (No Response)

Reviewer #2: In my opinion manuscript may be accepted for publication.

Reviewer #3: (No Response)

**Summary and General Comments**

Reviewer #1: (No Response)

Reviewer #2: The manuscript entitled "Absence of anti-Babesia microti antibody in commercial Intravenous immunoglobulin (IVIG)” deals with a very important aspect of the therapeutic effect of IVIG as adjunctive therapy in babesiosis treatment, especially in immunocompromised patients. Overall this is a well-conceived, relevant and rigorously implemented study, and I believe the results contained would further knowledge in this area. Therefore, I congratulate the authors on well-conducted research and valuable results. However, I have one comment to clarify – why were only a portion of IVIG samples tested using ELISA, while 57 commercial samples were tested using IFA? I think this short explanation should be added to the text.

Reviewer #3: This manuscript demonstrated the anti-Babesia microti antibody titers of commercial IVIG and can provide useful insight into the potential therapeutic efficacy of IVIG for babesiosis.

However, due to the several defects, it was determined that this manuscript was insufficient to publish.

1) Lack of information about a 66-year-old male donor

The donor information used for the data in Fig. 1 should be explained in the Materials and Methods section, not in the Results section.

There was no information on whether informed consent was obtained from this donor.

The information on the IVIG administered to this donor should be presented, along with a detailed list of the 57 samples tested in this study.

Authors stated that this donor had B-cell lymphoma, but did not write when he received rituximab.

The description of rituximab in the Discussion section should be presented in the Introduction section. In the Discussion section, the authors should discuss the results of the donor's serum IgG levels in relation to the history of rituximab administration, IVIG administration, etc.

Since clindamycin was administered at the same time as the IVIG in this study, the efficacy of IVIG should be discussed in considering of antibiotics, and should be compared with the results with antibiotics alone.

In addition, it would be interesting to determine if anti-Babesia microti antibodies are present in the serum of this recovered donor.

2) Lack of detailed information about IVIG

More detailed information on all 57 IVIG samples, including name, lot, and collection site, should be presented in a supplemental table.

3) No IFA results

There were no figures or discussion of the IFA results for the 57 samples of IVIG.

For IFA, a comparison with the results of non-infected RBCs as controls is needed, but was not stated.

4) ELISA problems

The product name of ELISA assay kit should be presented.

Even though the manufacturer's positive control was used, the PC results in Fig. 2 showed a large difference (> 1.0 OD) between the two replicates. It should be re-tested.

PLOS authors have the option to publish the peer review history of their article (what does this mean?). If published, this will include your full peer review and any attached files.

Reviewer #1: No

Reviewer #2: No

Reviewer #3: No
---

## [Decision Letter · Decision Letter 1]

1 Feb 2024

Dear Dr. Kostka,

Thank you very much for submitting your manuscript "Absence of Anti-<i>Babesia microti<i> Antibody in Commercial Intravenous Immunoglobulin (IVIG)" for consideration at PLOS Neglected Tropical Diseases. As with all papers reviewed by the journal, your manuscript was reviewed by members of the editorial board and by several independent reviewers. The reviewers appreciated the attention to an important topic. Based on the reviews, we are likely to accept this manuscript for publication, providing that you modify the manuscript according to the review recommendations. 

The manuscript has been improved according to the reviewer's suggestions. Some minor revisions are still required.

Sincerely,

Angela Monica Ionica, Ph.D.

Academic Editor

Abhay Satoskar

Section Editor

The manuscript has been improved according to the reviewer's suggestions. Some minor revisions are still required.

Reviewer's Responses to Questions

**Key Review Criteria Required for Acceptance?**

**Methods**

-Are the objectives of the study clearly articulated with a clear testable hypothesis stated?

-Is the study design appropriate to address the stated objectives?

-Is the population clearly described and appropriate for the hypothesis being tested?

-Is the sample size sufficient to ensure adequate power to address the hypothesis being tested?

-Were correct statistical analysis used to support conclusions?

-Are there concerns about ethical or regulatory requirements being met?

Reviewer #1: (No Response)

Reviewer #2: I have no comments.

Reviewer #3: Need to check the number of IVIG samples again in the full text. The number of IVIG lots appears to be 58 in Table S1, not 57 (line 108). In addition, was the data for 24 (or 28? according to the response to Reviewer 2) additional IVIG samples added?

**Results**

-Does the analysis presented match the analysis plan?

-Are the results clearly and completely presented?

-Are the figures (Tables, Images) of sufficient quality for clarity?

Reviewer #1: (No Response)

Reviewer #2: I have no comments.

Reviewer #3: In the Methods section, ELISA was explained after IFA, so the same order should be used in the Results section (lines 159-176).

**Conclusions**

-Are the conclusions supported by the data presented?

-Are the limitations of analysis clearly described?

-Do the authors discuss how these data can be helpful to advance our understanding of the topic under study?

-Is public health relevance addressed?

Reviewer #1: (No Response)

Reviewer #2: I have no comments.

Reviewer #3: (No Response)

**Editorial and Data Presentation Modifications?**

Reviewer #1: (No Response)

Reviewer #2: I have no comments.

Reviewer #3: Regarding the duplicate statement that rituximab depletes B cells in lines 73 and 77, delete one of them.

The differences between A, B, and C attached to the sample numbers in Table S1 should be explained in the legend. Alternatively, they should be described as one lot with one sample number. 

If possible, the results of IFA and the ELISA OD450 value used for Fig 2 should be added in Table S1. This would make it easier for the reader to understand which samples were tested in IFA and ELISA.

**Summary and General Comments**

Reviewer #1: (No Response)

Reviewer #2: All comments of both reviewers were carefully considered and answered step by step. In my opinion, the manuscript is now in a form to be published.

Reviewer #3: (No Response)

PLOS authors have the option to publish the peer review history of their article (what does this mean?). If published, this will include your full peer review and any attached files.

Reviewer #1: No

Reviewer #2: No

Reviewer #3: No

Figure Files:

Data Requirements:

Reproducibility:

References

---

## [Editor Report · Decision Letter 2]

28 Feb 2024

Dear Dr. Kostka,

We are pleased to inform you that your manuscript 'Absence of Anti-*Babesia microti* Antibody in Commercial Intravenous Immunoglobulin (IVIG)' has been provisionally accepted for publication in PLOS Neglected Tropical Diseases.**

*Before your manuscript can be formally accepted you will need to complete some formatting changes, which you will receive in a follow up email. A member of our team will be in touch with a set of requests.*

*Please note that your manuscript will not be scheduled for publication until you have made the required changes, so a swift response is appreciated.*

*IMPORTANT: The editorial review process is now complete. PLOS will only permit corrections to spelling, formatting or significant scientific errors from this point onwards. Requests for major changes, or any which affect the scientific understanding of your work, will cause delays to the publication date of your manuscript.*

*Should you, your institution's press office or the journal office choose to press release your paper, you will automatically be opted out of early publication. We ask that you notify us now if you or your institution is planning to press release the article. All press must be co-ordinated with PLOS.*

*Thank you again for supporting Open Access publishing; we are looking forward to publishing your work in PLOS Neglected Tropical Diseases.*

*Best regards,*

Angela Monica Ionica, Ph.D.

Academic Editor

Abhay Satoskar

Section Editor

---

## [Editor Report · Acceptance letter]

8 Mar 2024

Dear Dr Kostka,

We are delighted to inform you that your manuscript, "Absence of Anti-<i>Babesia microti<i> Antibody in Commercial Intravenous Immunoglobulin (IVIG)," has been formally accepted for publication in PLOS Neglected Tropical Diseases.

Best regards,

Shaden Kamhawi

co-Editor-in-Chief

Paul Brindley

co-Editor-in-Chief
